# Profiles of Cytokines Secreted by ARPE-19 Cells Exposed to Light and Incubated with Anti-VEGF Antibody

**DOI:** 10.3390/biomedicines9101333

**Published:** 2021-09-27

**Authors:** Tomohito Sato, Masaru Takeuchi, Yoko Karasawa, Masataka Ito

**Affiliations:** 1Department of Ophthalmology, National Defense Medical College, Tokorozawa 359-8513, Japan; dr21043@ndmc.ac.jp (T.S.); kyop518@ndmc.ac.jp (Y.K.); 2Department of Developmental Anatomy and Regenerative Biology, National Defense Medical College, Tokorozawa 359-8513, Japan; masataka@ndmc.ac.jp

**Keywords:** cytokine, ARPE-19 cell, light irradiation, anti-VEGF antibody

## Abstract

The retinal pigment epithelium (RPE) is the major source of cytokines in the retina regulating the intraocular immune environment, and a primary target of photodamage. Here, we examined 27 types of cytokines secreted by ARPE-19 cells exposed to visible light and incubated with aflibercept or ranibizumab, which are two anti-vascular endothelial growth factor (VEGF) antibodies. The cells were cultured for 24 h in the dark or under 2000 lux irradiation from a daylight-colored fluorescent lamp, and cytokine levels in the culture supernatant were measured. In the light-irradiated culture, the levels of IL-9, IL-17A and bFGF were higher, and the levels of IL-6, IL-7, IL-8 and MCP-1 were lower than those in the dark culture, while there was no significant difference with the VEGF-A level. In subgroup analyses of the light-irradiated culture, the bFGF level under 250 to 2000 lux irradiation was elevated in a light intensity-dependent manner. In culture exposed to blue, green or red light, the bFGF level was elevated by blue light and was high compared to that by green or red light. In culture with aflibercept or ranibizumab in the dark, the levels of IL-6, IL-8, bFGF and MCP-1 were increased, and the IL-12 level decreased synchronously with a reduction in the VEGF-A level. Our findings indicate that continuous irradiation of visible light and VEGF suppression may be an influential factor in expression patterns of inflammatory cytokines secreted by human RPE cells.

## 1. Introduction

The retinal pigment epithelium (RPE) is the outermost monolayer composed of pigmented cells that form the blood–retina barrier (BRB) limiting access of blood cells and proteins to the retina [1]. The BRB is classified into two components: the inner BRB and the outer BRB [2]. The breakdown of one or both of the BRBs is involved in the etiologies of some retinal diseases [2], and the outer BRB consists of the RPE, Bruch’s membrane and the choriocapillaris [3]. The outer BRB is associated with a major pathological lesion in age-related macular degeneration (AMD) [2]. On the other hand, the RPE regulates immunological responses and maintains an immune-privileged status by secreting immunosuppressive factors in the retina and choroid [4]. Therefore, it is assumed that dysfunction of the RPE is a potential risk factor for developing several retinal diseases such as retinitis pigmentosa [5] and AMD [6].

AMD is the leading cause of blindness among the elderly in developed countries and is characterized by degeneration of the RPE and photoreceptor cells in the macula [7,8]. AMD is divided into two forms according to clinical features: dry AMD and neovascular AMD (nAMD) [9]. It is considered that low-grade inflammation, oxidative stress, genetics, etc., are involved in the pathology of AMD [10,11,12]. At present, the anti-vascular endothelial growth factor (VEGF) agent is used as the first-line treatment for nAMD and is also applied to other retinal diseases such as diabetic macular edema, macular edema secondary to retinal vein occlusion and myopic choroidal neovascularization (mCNV) [13].

The retina is a light-sensitive tissue composed of the RPE and neuroretinal components including photoreceptor cells [14]. The RPE is frequently exposed to oxidative stress induced by light exposure in modern life, and the oxidative stress generates reactive oxygen species (ROS), which is a risk factor for the development of AMD [15,16,17]. In the Beaver Dam Eye Study, sunlight exposure was verified as a potential aggravating factor for the progression to nAMD and late AMD [18]. Therefore, basic exploratory research to examine the complicated biochemical effects of light exposure, especially blue light [17] and anti-VEGF agents in the RPE, is required for a better understanding of the pathophysiology of AMD.

Cytokines are small, nonstructural proteins secreted by a variety of immune cells such as macrophages, and nonimmune cells including RPE cells and fibroblasts [19]. Cytokines play a key role in inflammation and induce various immune responses such as the following: stimulation or inhibition of cell proliferation, cytotoxicity/apoptosis, antiviral activity, cell growth and differentiation [19,20]. In addition, cytokines demonstrate multiple and diverse functions simultaneously by interacting with each other [20,21]. On the other hand, RPE cells produce a wide variety of cytokines that activate the resident ocular cells and attract leukocytes under inflammatory conditions [4]. RPE cells are also highly sensitive to numerous cytokines [4]. Therefore, to reveal the complicated effects of cytokines secreted by RPE cells in various situations, the method of comprehensive evaluation using the multivariate analysis technique could be useful.

In the present study, we showed the profiles of inflammatory cytokines secreted by ARPE-19 cells as a human RPE cell line under in vitro conditions which attempted to mimic clinical situations as follows: (1) continuous light irradiation of a fluorescent lamp as an imitated ambient light irradiation, and (2) exposure to aflibercept or ranibizumab, which are two anti-VEGF antibodies used for the treatment of nAMD.

## 2. Methods

### 2.1. Cell Culture and Medium for Illuminated Culture

The human RPE cell line ARPE-19 was purchased from the American Type Culture Collection (Manassas, VA, USA, Cat #: 2302). The cells were cultured in DMEM/F12 (Sigma-Aldrich, Poole, UK) supplemented with 10% fetal bovine serum (FBS; JRH Biosciences, Lenexa, KS, USA) and antibiotics (100 U/mL penicillin G and 100 μg/mL streptomycin sulfate; Sigma-Aldrich) at 37 °C in a humidified atmosphere of 5% CO_2_ and 95% air [22,23,24]. The cells were trypsinized, and 1.5 × 10^4^ cells per well were subcultured in 4-well plates (Thermo Scientific, Nunc, Rochester, NY, USA), in which the area was 1.9 cm^2^. The cells were grown to confluence for 7 days. To eliminate shielding effects of coloration in the culture medium under continuous light irradiation from a fluorescent lamp, we prepared a colorless PBS-based medium consisting of Dulbecco’s phosphate-buffered saline (PBS; Sigma-Aldrich) supplemented with 1% FBS, 1 mg/mL glucose, 0.1 mg/mL CaCl_2_, 46.8 μg/mL MgCl_2_ and the antibiotics [22,23,24]. The cells were cultured for 24 h in the dark or under light irradiation, and the levels of cytokines in the culture supernatant were measured.

### 2.2. Cell Culture under Continuous Light Irradiation from Fluorescent Lamp

ARPE-19 cells maintained in confluent monolayers were used in our experiments. The monolayers were washed once with PBS and then cultured in the PBS-based medium in the dark within a metal box, or under 2000 lux irradiation from a daylight-colored fluorescent lamp (6500 K, Sunline; Hitachi, Tokyo, Japan) in an incubator (CPO_2_-171, Hirasawa, Tokyo, Japan) fitted with the lamp. Neutral-density filters (Nikon, Tokyo, Japan) were used to reduce the light intensity from 2000 to 1000, 500 or 250 lux for testing biochemical effects of light exposures at various intensities.

For blocking specific light wavelengths, the cells were cultured in the 4-well plates. A blue, green or red dichroic mirror (DM; 4 × 4 cm^2^, Koshin, Tokyo, Japan) was attached on top of the culture plate [22,24]. Each DM reflects light wavelengths corresponding to each color and allows light wavelengths of other colors to pass through. The sides of the plates were covered with aluminum foil to prevent transmission of light from the fluorescent lamp.

### 2.3. Cell Culture Incubated with Anti-VEGF Antibody

Aflibercept (Regeneron Pharmaceuticals, Inc., Tarrytown, NY, USA) and ranibizumab (Genentech, Inc., South San Francisco, CA, USA) were used as anti-human VEGF antibodies. The concentration of aflibercept or ranibizumab in the adult human vitreous cavity immediately after intravitreal injection in clinical setting is estimated as follows: 0.05 mL (volume of injection) × 40 mg/mL (dose of aflibercept) or 10 mg/mL (dose of ranibizumab)/4 mL (estimated volume of the vitreous cavity) = 0.50 mg/mL of aflibercept or 0.125 mg/mL of ranibizumab. The monolayers were washed once with PBS and then cultured in PBS-based medium containing 0.50 mg/mL aflibercept (estimated neat concentration in the vitreous cavity), or 0.125 mg/mL (1/4 dilution), 31.3 μg/mL (1/16), 7.81 μg/mL (1/64) or 0 μg/mL as a relative control without reagents of aflibercept in the dark or under light irradiation. In the same manner, the cells were incubated in PBS-based medium containing 0.125 mg/mL ranibizumab (neat), or 31.3 μg/mL (1/4 dilution), 7.81 μg/mL (1/16), 1.95 μg/mL (1/64) or 0 μg/mL as a relative control without reagents of ranibizumab.

### 2.4. Cytokine Measurement

The culture supernatants of ARPE-19 cells incubated for 24 h were collected in sterile tubes and stored at −80 °C until assay. Twenty-seven types of cytokines in the culture supernatants were measured using the Bio-Plex Human Cytokine 27-Plex panel (Bio-Rad, Hercules, CA, USA) with a multiplex bead analysis system (Bio-Plex Suspension Array System; Bio-Rad) according to the manufacturer’s instructions. All standards and samples were assayed in duplicate. Cytokine levels below detectable levels were regarded as 0 μg/mL for statistical analysis [25,26].

Some fluctuations of secreted cytokine levels occur, even if the experiments are strictly performed. In this study, preliminary experiments were conducted twice, and the results obtained in the third experiment were adopted in this study. The results in the preliminary experiments were almost the same as those in the third experiment.

### 2.5. Statistical Analysis

Statistical analyses were performed using the statistical add-in software for Excel (BellCurve for Excel^®^ version 3.21, SSRI Co., Ltd., Tokyo, Japan). Data are expressed as mean ± standard deviation. The unpaired Student t-test was used for parametric comparison between two groups. Dunnett’s test followed by post hoc non-repeated measures ANOVA was used for parametric comparisons with control, and the Tukey–Kramer test was adopted for all parametric pairwise comparisons. Pearson’s correlation was used for examining parametric correlations between 2 groups. Inflammatory cytokines with detection rates of 80% or more and high concentrations greater than 10 pg/mL were adopted in Pearson’s correlation test, hierarchical cluster analysis and principal component analysis (PCA). Hierarchical cluster analysis was performed using Ward’s method with Euclidean distance as the distance metric [25,26]. A two-tailed test was applied to the univariate analysis used. A *p* level less than 0.05 was considered to be statistically significant.

## 3. Results

### 3.1. Inflammatory Cytokine Levels in ARPE-19 Cell Culture Supernatant

The levels of inflammatory cytokines in the culture supernatant of ARPE-19 cells incubated for 24 h in the dark or under continuous 2000 lux irradiation of visible light from a fluorescent lamp with a correlated color temperature of 6504 K (light irradiation) are shown in Table 1. The inflammatory cytokines with detection rates of 80% or more were IL-6, IL-7, IL-8, IL-9, IL-10, IL-12, IL-15, IL-17A, bFGF, G-CSF, IP-10, MCP-1 and VEGF-A in the dark culture and the light-irradiated culture. The levels of IL-9, IL-17A and bFGF were higher, and the levels of IL-6, IL-7, IL-8 and MCP-1 were lower in the light-irradiated culture than those in the dark culture, while there was no significant difference with the levels of the remaining cytokines including VEGF-A. To evaluate the comprehensive effects of the secreted cytokines by hierarchical cluster analysis and PCA, the cytokines adopted as explanatory variables in the analyses should be reliably detectable, and those levels are required to be reasonably high. Furthermore, the number of samples used in the analyses must be one or more than that of the explanatory variables. IL-6, IL-8, IL-12, IL-17A, bFGF, IP-10, MCP-1 and VEGF-A were inflammatory cytokines with detection rates of 80% or more and high concentrations greater than 10 pg/mL (Table 1). Therefore, we used those cytokines as appropriate explanatory variables in subsequent analyses.

### 3.2. Correlation between Inflammatory Cytokine Levels

The correlation between inflammatory cytokine levels in the culture supernatant of the cells incubated for 24 h in the dark or under light irradiation is shown in Figure 1. In the dark culture, the IL-6 level correlated positively with the MCP-1 level. Furthermore, there was a positive correlation between the bFGF and IP-10 levels (Figure 1A,C,E). In the light-irradiated culture, positive correlations were detected between the IL-6 and IL-8 levels, between the L-8 and bFGF levels, between the IL-12 and VEGF levels and between the IL-17A and IP-10 levels (Figure 1B,D,F). A negative correlation was revealed between the IL-6 and bFGF levels, and between the IP-10 and VEGF levels. The results of the correlation analysis indicate that the light irradiation changed the specific interrelationships between the IL-6 and MCP-1 levels, and between the bFGF and IP-10 levels in the dark culture, and created new special interrelationships between other inflammatory cytokine levels.

### 3.3. Hierarchical Cluster Analysis of Inflammatory Cytokine Expression Patterns

Cluster analysis was performed to classify inflammatory cytokines into relative similar groups called clusters [27]. In both the dark and light-irradiated cultures, the cytokines were roughly classified into a single cytokine with a high expression level, and a large cytokine group with similar properties, as follows: (1) VEGF-A, and (2) a group composed of the remaining cytokines (Figure 2). Furthermore, the group was broadly divided into MCP-1 with a relatively high expression level, and a cytokine cluster consisting of IL-6, IL-8, IL-12, IL-17A, bFGF and IP-10. The results of the hierarchical cluster analysis show that the light irradiation had almost no influence on the fundamental properties of the inflammatory cytokines secreted by the cells.

### 3.4. Principal Component Analysis of Inflammatory Cytokine Expression Patterns

PCA was performed to present multivariate data simply by converting a set of many correlated cytokines into a set of fewer uncorrelated cytokines [28]. In the dark culture, the eigenvalues for the first principal component (PC1), second principal component (PC2) and third principal component (PC3) were 3.25, 1.95 and 1.18, respectively (Figure 3G). The contribution ratios of PC1, PC2 and PC3 in the variance of all data were 40.6%, 24.4% and 14.7%, respectively. In principal component loading (PCL) analysis of PC1, VEGF-A, IL-12, IL-6 and IL-17A formed a cytokine group, while IP-10 was located on the opposite side of the group (Figure 3D). Based on the positive or negative PCL sizes of those cytokines, the primary effect of PC1 was presumed to be an angiogenesis-related function. In PCL analysis of PC2, all of the cytokines had positive PCLs, and bFGF showed the largest PCL size (Figure 3E). Therefore, it was assumed that the predominant effect of PC2 was an angiogenesis-related function with inflammation. In PCL analysis of PC3, MCP-1 and IL-6 formed an inflammatory cytokine cluster, but IL-12 was located on the opposite side of the cluster (Figure 3F). Based on the arrangements of those cytokines, the principal effect of PC3 was estimated to be an inflammatory-related function.

### 3.5. Principal Component Analysis of Inflammatory Cytokine Expression Patterns under Light Irradiation

In the light-irradiated cultures, the eigenvalues of PC1, PC2 and PC3 were 2.77, 2.21 and 1.50, respectively (Figure 3N). The contribution ratios of PC1, PC2 and PC3 were 34.6%, 27.6% and 18.8%, respectively. In PCL analysis of PC1, bFGF, IL-17A, IP-10 and IL-8 formed a cytokine group, while MCP-1 and IL-6 were located on the opposite side of the group (Figure 3K). Based on the PCL sizes and the arrangements of those cytokines, it was assumed that the primary effect of PC1 was an angiogenesis-related function accompanied by inflammation. In PCL analysis of PC2, VEGF-A, IL-12 and IL-6 formed a cytokine cluster, while IP-10 was located on the opposite side of the cluster (Figure 3L). Based on the arrangements of the cluster and IP-10, the main effect of PC2 was presumed to be an angiogenesis-related function, similar to that of PC1 in the dark culture. In PCL analysis of PC3, a cytokine group consisted of MCP-1 and IL-6, while another group composed of VEGF and IL-12 was located on the opposite side of the group (Figure 3M). It was supposed that the dominant effect of PC3 was an inflammatory-related function in competition with an angiogenesis-related function.

### 3.6. Cytokine Levels under Various Light Intensities

In subgroup analysis of the light-irradiated culture, to reveal the changes in cytokine levels depending on the light intensity, it was reasonable that the cytokines with significant differences in levels between the dark culture and the light-irradiated culture were selected. Therefore, IL-6, IL-8, IL-17A, bFGF and MCP-1, excluding IL-12 and VEGF-A (see Table 1), were chosen as appropriate cytokines in further research. The levels of inflammatory cytokines in the culture supernatant of the cells incubated for 24 h in the dark or under light irradiation with 250 lux, 500 lux, 1000 lux or 2000 lux are shown in Figure 4. Under light irradiation, the bFGF level was elevated in a light intensity-dependent manner (Figure 4E). On the other hand, the levels of IL-6, IL-8 and MCP-1 were almost the same among the cultures exposed to 250 lux, 500 lux, 1000 lux and 2000 lux (Figure 4B,C,F). The results of the subgroup analysis show that the inflammatory cytokines whose levels were changed in a light intensity-dependent manner or not were mixed together. Therefore, the cytokine responses to light exposure in the cells fluctuated according to the light irradiation conditions and were unequal.

### 3.7. Cytokine Levels under Blue, Green or Red Light Irradiation

In another subgroup analysis of the light-irradiated culture, the cells were cultured for 24 h in the dark or under the irradiation of blue, green, red or full-spectrum light (whole light) from the fluorescent lamp. The levels of IL-6, IL-8, IL-17A, bFGF and MCP-1 in the culture supernatant of the dark culture or the light-irradiated culture with the indicated light wavelengths are shown in Figure 5. The levels of IL-6 and IL-8 under the irradiation of blue, green or red light were lower than those in the dark (Figure 5B,C). There was no significant difference in the levels of IL-6, IL-8 and MCP-1 among the cultures exposed to blue, green, red or whole light (Figure 5B,C,F,G). On the other hand, the levels of IL-17A and bFGF in the culture exposed to blue light were high compared with those exposed to red or whole light (Figure 5D,E,G–I). Furthermore, the bFGF level under blue light was higher than that under green light. The results of the subgroup analysis indicate that the inflammatory cytokines whose levels fluctuated depending on the exposure to specific light wavelengths or not were present simultaneously. Therefore, the cytokine responses in the cells could be changed in accordance with specific light wavelengths.

### 3.8. Levels of Inflammatory Cytokines with Anti-VEGF Antibody

The levels of inflammatory cytokines in the culture supernatant of the cells incubated for 24 h in the dark with 0.50 mg/mL aflibercept, 0.125 mg/mL ranibizumab or neither as a relative control without reagents of aflibercept or ranibizumab are shown in Table 2. The levels of IL-6, IL-8, IL-12, L-17A, bFGF, MCP-1 and VEGF-A were higher than 10 pg/mL in at least one of the three cultures and were reasonably high. Therefore, we chose those cytokines as appropriate explanatory variables in the subsequent subgroup analyses. In the dark culture with aflibercept or ranibizumab, the levels of IL-6, IL-8, IL-17A, bFGF and MCP-1 were higher than those in the control, while the IL-12 and VEGF-A levels were lower. The levels of IL-12 and VEGF-A in the culture with ranibizumab were high compared to those with aflibercept. The results suggest that the elevated cytokines could work complementarily as an angiogenesis-related substance under VEGF suppression.

### 3.9. Levels of Inflammatory Cytokines with Anti-VEGF Antibody under Light Irradiation 

The levels of inflammatory cytokines in the supernatant of the cells incubated for 24 h under light irradiation with 0.50 mg/mL of aflibercept, 0.125 mg/mL of ranibizumab or neither as the relative control are described in Table 3. The levels of IL-12, MCP-1 and VEGF-A were more than 10 pg/mL in at least one of the three cultures. In the light-irradiated cultures with aflibercept or ranibizumab, the MCP-1 level was higher than that in the control, while the IL-12 and VEGF-A levels were lower. Furthermore, the VEGF-A level in the light-irradiated culture with ranibizumab was higher than that with aflibercept. The results are similar to those in the dark culture with aflibercept or ranibizumab.

### 3.10. Cytokine Levels in Aflibercept- or Ranibizumab-exposed ARPE-19 Cells Cultured in Dark and under Light Irradiation

Table 4 shows comparisons of the cytokine levels between the dark and the light-irradiated cultures with 0.50 mg/mL aflibercept, or the dark and light-irradiated cultures with 0.125 mg/mL ranibizumab. The levels of IL-1ra, IL-6, IL-8, IL-17A, bFGF, MCP-1 and VEGF-A in the dark culture with aflibercept or ranibizumab were more than 10 pg/mL and were selected as appropriate cytokines for the comparisons. In both cultures incubated with aflibercept and ranibizumab, the levels of IL-6, IL-8, bFGF and MCP-1 under light irradiation were lower than those in the dark. The IL-1ra level in the light-irradiated culture incubated with aflibercept was elevated compared to that in the dark culture. On the other hand, there was no significant difference with the VEGF-A level between the cultures incubated with aflibercept in the dark and under light irradiation because the VEGF-A levels were extremely low in both cultures (dark: 3.24 pg/mL; light: 3.15 pg/mL). In the cultures incubated with ranibizumab, the VEGF-A level under light irradiation was lower than that in the dark (dark: 115.2 pg/mL; light: 47.4 pg/mL). The results indicate that the light irradiation decreased the levels of cytokines secreted by the cells incubated with anti-VEGF antibody, almost uniformly.

### 3.11. Cytokine Levels in Dark Culture Incubated with Aflibercept at Various Concentrations

The levels of inflammatory cytokines in the culture supernatant of the cells incubated with 7.81 μg/mL, 31.3 μg/mL, 0.125 mg/mL and 0.50 mg/mL aflibercept or no concentration as the relative control in the dark for 24 h are shown in Figure 6. The levels of IL-6, IL-8 and bFGF were increased in a dose-dependent manner (Figure 6B,C,F). The MCP-1 level elevated uniformly in all cultures incubated with 7.81 μg/mL aflibercept and above (Figure 6G). On the other hand, the IL-12 and VEGF-A levels were remarkably reduced to almost zero in the cultures incubated with 31.3 μg/mL aflibercept and above (Figure 6D,H). Multiple comparisons of cytokine levels among the dark cultures incubated with aflibercept at the indicated concentrations are described in Appendix A. The results of the subgroup analysis show that the inflammatory cytokines whose levels were increased or decreased in a dose-dependent manner were mixed together, while the MCP-1 level elevated even at low concentrations. Therefore, the cytokine responses in the cells incubated with aflibercept would differ and be changed according to the individual cytokines.

### 3.12. Cytokine Levels in Light-Irradiated Culture Incubated with Aflibercept at Various Concentrations

The levels of inflammatory cytokines in the culture supernatant of the cells incubated with 7.81 μg/mL, 31.3 μg/mL, 0.125 mg/mL and 0.50 mg/mL aflibercept or no concentration as the relative control under light irradiation for 24 h are also shown in Figure 6. The levels of IL-6 and MCP-1 were elevated in the light-irradiated cultures incubated with 0.50 mg/mL aflibercept (Figure 6J,O), while the IL-12 and VEGF-A levels were significantly decreased to nearly zero at 7.81 μg/mL aflibercept and above (Figure 6L,P). Multiple comparisons of cytokine levels among the light-irradiated cultures incubated with aflibercept at the indicated concentrations are described in Appendix A. The results of the subgroup analysis indicate that the light irradiation maintained the fluctuations in the cytokine levels in the dark culture incubated with aflibercept but attenuated the degrees of the fluctuations.

### 3.13. Cytokine Levels in Dark Culture Incubated with Ranibizumab at Various Concentrations

Figure 7 shows the levels of inflammatory cytokines in the culture supernatant of the cells incubated with 1.95 μg/mL, 7.81 μg/mL, 31.3 μg/mL and 0.125 mg/mL ranibizumab or no concentration as the relative control in the dark for 24 h. The levels of IL-6, IL-8 and bFGF were increased in a dose-dependent manner (Figure 7B, C and F). The MCP-1 level elevated uniformly in all cultures incubated with 7.81 μg/mL ranibizumab and above (Figure 7G). On the other hand, the IL-12 and VEGF-A levels were increased at 7.81 μg/mL ranibizumab but decreased thereafter in a dose-dependent manner (Figure 7D,H). Multiple comparisons of cytokine levels among the dark cultures incubated with ranibizumab at the indicated concentrations are described in Appendix A. The results of the subgroup analysis demonstrate that the fluctuations in the cytokine levels in the dark culture incubated with ranibizumab were approximately the same as those with aflibercept.

### 3.14. Cytokine Levels under Light Irradiation Culture Incubated with Ranibizumab at Various Concentrations

The levels of inflammatory cytokines in the culture supernatant of the cells incubated with 1.95 μg/mL, 7.81 μg/mL, 31.3 μg/mL and 0.125 mg/mL ranibizumab or no concentration as the relative control under light irradiation for 24 h are also shown in Figure 7. The MCP-1 level was increased in the light-irradiated cultures incubated with 0.125 mg/mL ranibizumab (Figure 7O), while the levels of IL-12 and VEGF-A were decreased in a dose-dependent manner (Figure 7L,P). Multiple comparisons of cytokine levels among the light-irradiated cultures incubated with ranibizumab at the indicated concentrations are described in Appendix A. The results of the subgroup analysis indicate that the light irradiation sustained the changes in the cytokine levels in the dark culture incubated with ranibizumab, and that the changes in the light-irradiated cultures incubated with ranibizumab were quite similar to those with aflibercept.

## 4. Discussion

The present study was designed to investigate the expression patterns of inflammatory cytokines secreted by ARPE-19 cells under various in vitro conditions, which attempted to mimic the clinical situation. Our experimental results yield several noteworthy findings as follows: (1) IL-6, IL-8, IL-12, IL-17A, bFGF, IP-10, MCP-1 and VEGF-A were the primary cytokines secreted by the cells, and the comprehensive effect of the secreted cytokines was presumed to be an angiogenesis-related function by PCA. (2) Mild continuous irradiation of visible light suppressed the secretion of most inflammatory cytokines by the cells, but VEGF-A production was maintained. bFGF expression was elevated in a light intensity-dependent manner and was also increased under irradiation of blue light compared to green or red light. (3) In the culture incubated with aflibercept or ranibizumab, the secretions of IL-6, IL-8, bFGF and MCP-1 were elevated, and IL-12 expression was decreased synchronously with the reduction in the VEGF-A level.

ARPE-19 cells are a spontaneously arising human RPE cell line with normal karyology established by Dunn et al. [29] and are widely used as an alternative to native human RPE cells [30]. We previously reported a mild photooxidative stress in ARPE-19 cells using our culture system [24] and showed no obvious cell damage after 24-h culture under continuous light irradiation of a fluorescent lamp [23,24] as an irradiation of ambient light. In this study, IL-6, IL-8, IL-12, IL-17A, bFGF, IP-10, MCP-1 and VEGF-A were detected as principal cytokines secreted by ARPE-19 cells (Table 1). In the PCA, the cumulative contribution ratio of PC1 and PC2 was approximately 65%, and the two components accounted for the majority of the variance in all data (Figure 3). PC1 and PC2 were dominantly composed of angiogenic-related cytokines including VEGF-A and bFGF, whose levels were sufficiently high to demonstrate cytokine effects (Table 1). VEGF-A is a critical signal protein that promotes angiogenesis and vascular leakage [31]. Basic FGF has biological effects on cell growth, differentiation, regeneration, neovascularization, senescence, etc. [32,33]. Therefore, we suppose that the comprehensive effect of cytokines secreted by ARPE-19 cells may be an angiogenic-related function. 

The RPE plays a role in ocular immune privilege, which regulates the production of anti-inflammatory cytokines and mediates antigen-specific regulatory immune activation [34,35,36]. In the PCA, PC3 accounted for around 15% of the variance in all data (Figure 3) and was predominantly composed of inflammatory-related cytokines such as MCP-1 and IL-12. The levels of MCP-1 and IL-12 secreted by the cells were high enough to exhibit cytokine effects (Table 1). MCP-1 is one of the key chemokines regulating the migration and infiltration of monocytes/macrophages [37]. MCP-1 works as an indirect angiogenesis inducer by recruiting macrophages, which, in turn, act as a source of angiogenic cytokines [38,39]. On the other hand, IL-12 is associated with the development of the T helper 1 cell response [40] and plays a central role in coordination with the innate and adaptive immunities [41]. Furthermore, IL-12 works as a strong anti-angiogenic cytokine in vivo [42]. Based on previous studies and our results, it is assumed that the RPE could modulate the intraocular immune environment by secreting inflammatory cytokines, which simultaneously contribute to the maintenance of the microvasculature in the retina and choroid.

In modern life, light exposure inevitably causes photooxidative stress to the retina [17]. As for the effects of oxidative stress and light exposure on cytokine production in human RPE cells, AnandBabu et al. [7] showed that the levels of IL-6, IL-8 and VEGF secreted by ARPE-19 cells incubated with 200 μM H_2_O_2_ for 24 h were elevated, and proapoptotic changes were present. Shen et al. [43] reported that 7378 K light-emitting diode (LED) illumination for 24 h, but not 2954 K, increased the levels of IL-6, IL-8 and VEGF-A and decreased the MCP-1 level in the culture supernatant of primary human RPE cells. In our study, continuous 2000 lux irradiation of a 6500 K fluorescent lamp was adopted as an experimental condition of light exposure, which induced no significant cellular damage in ARPE-19 cells incubated for 24 h [23,24]. Therefore, the expression patterns of cytokines secreted by human RPE cells may fluctuate depending on individual experimental conditions according to the cell type, light exposure method and degree of oxidative stress.

In the present study, the blue light irradiation elevated the expression of IL-17A and bFGF by ARPE-19 cells compared to the green or red light (Figure 5). ARPE-19 cells contain a lot of mitochondria, and cytochromes in the respiratory chain of mitochondria absorb light wavelengths within the 410–440 nm range [44]. Previous studies showed that blue/violet (400–450 nm) light exposure had an adverse effect on mitochondrial functions [45,46,47,48]. In brief, blue light exposure induced a reduction in adenosine triphosphate formation and an elevation in ROS production, resulting in damage to lipids, proteins and deoxyribonucleic acids in cells with no melanin pigmentation [45,46,47,48]. Furthermore, the blue light (approximately 400–500 nm) exposure caused cell death in the RPE of an albino rat at a light intensity one tenth of that required with green light [49]. In contrast, it is assumed that red light (about 600–1000 nm) exposure induces no cytotoxic effect on cells with no melanin pigmentation, and that ROS could be absorbed by photon absorption stimulation of the mitochondrial enzyme cytochrome c oxidase via red light [50,51,52]. Therefore, the changes in mitochondrial function by red light exposure may have a potentially beneficial effect of attenuating oxidative stress, which results in suppressions of inflammation and cell death [51,53,54]. In this study, the elevation in the IL-17A level secreted by the cells was induced under blue light irradiation, but not under the red light. In the future, further studies are expected to investigate specific expression patterns of cytokines secreted by human RPE cells, which depend on individual experimental condition of light exposure.

In this study, the exposure to aflibercept or ranibizumab increased the secretions of IL-6, IL-8, bFGF and MCP-1 by ARPE-19 cells incubated in the dark and decreased IL-12 expression synchronously with a reduction in the VEGF-A level (Figure 6 and Figure 7). We previously reported the profile of aqueous humor (AH) cytokines in treatment-naïve nAMD eyes before and after intravitreal injection of aflibercept (IVA) [25]. In that report, the changes in the AH cytokine levels, especially the IL-12 and VEGF levels, before and after IVA in the nAMD eyes were quite similar to those of the cytokine levels in the culture supernatant of ARPE-19 cells incubated with aflibercept or not (Table 2). Saenz-de-Viteri et al. [55] showed no measurable cytotoxicity after single or repeated doses of aflibercept and ranibizumab under normal conditions and oxidative stress conditions exposed to H_2_O_2_ in ARPE-19 cells. Therefore, the present in vitro model could be an aid for examining the pathophysiology of nAMD and mCNV under anti-VEGF therapy because the primary etiology of retinal diseases is considered to be the dysfunction of the RPE.

As for the anti-VEGF antibodies used in this study, aflibercept is a recombinant fusion protein composed of VEGF-binding portions that mimics VEGF receptor-1 and -2 and the Fc region of human immunoglobulin G1 and binds VEGF-A and -B and placental growth factor with very high affinity [56]. Ranibizumab consists of a humanized monoclonal antibody fragment which binds VEGF-A with high affinity; in other words, it stops VEGF-A from binding to VEGF receptor-1 and -2 [57]. Therefore, the reagents of aflibercept are completely different to those of ranibizumab. Furthermore, the reagents contain biological components with potential influences on cell activity. In our experiments with anti-VEGF antibodies, the culture without reagents of the anti-VEGF antibodies was used as a relative control because the composition of those reagents is unpublished, and the reagent control cannot be established. Thus, prudence should be exercised when conducting comparisons of the biological effects among cultures incubated with aflibercept, ranibizumab or no reagent.

Melanin pigmentation is believed to protect the skin and the eyes against phototoxic reactions induced by ultraviolet or blue light [58,59,60]. In the eye, melanin acts as a cellular antioxidant and may protect RPE cells against oxidative stress elicited by visible light or by redox-active metal ions such as iron [60,61,62,63]. Melanin secreted by melanocytes is chemically classified into two types as follows: the insoluble, black to brown eumelanin, and the alkaline-soluble, yellow to reddish-brown pheomelanin [60,64,65]. It is supposed that eumelanin is photoprotective, and pheomelanin may be phototoxic to pigmented tissues [58,65]. Eumelanin is exclusively present in the RPE, while melanin in the skin is always a mixture of eumelanin and pheomelanin [66,67]. Regarding cell maturity, melanin pigmentation is an important phenotypic indicator in human RPE cells [68,69]. ARPE-19 cells will present evident melanin pigmentations by a long-term culture of more than four months [30]. In this study, we adopted a short-term culture of ARPE-19 cells, and the cells presented no melanin pigmentation during the culture period. Therefore, the maturity of the ARPE-19 cells used in our study may be morphologically insufficient, and melanin pigmentation might have almost nothing to do with the changes in the cytokine levels in the light-irradiated culture.

In senescence studies, the use of ARPE-19 cells has some limitations [70]. ARPE-19 subculture contains both mortal and immortal cells, and a large proportion of the cells are immortalized [71,72]. Unlike cells with a limited number of divisions, immortal cells do not undergo replicative senescence [70]; however, the cells are prone to stress-induced senescence [73,74]. Previous cell senescence studies indicated that the senescence of ARPE-19 cells was induced by hydroxyl peroxide in nontoxic concentrations [75,76,77,78]. In the light-irradiated culture of our study, mild photooxidative stress occurred and induced no significant cell damage in ARPE-19 cells [24]. Thus, discretion is required when ARPE-19 cells are used as a native human RPE cell model in the basic research of retinal diseases such as AMD [72].

The present study has several limitations as follows: (1) twenty-seven types of inflammatory cytokines were limitedly evaluated, and other candidate cytokines were not examined; (2) continuous 2000 lux irradiation of a daylight-colored fluorescent lamp for 24 h [23,24] was used as the condition of visible light exposure in our study, and other conditions of light exposure were not investigated; (3) in the experiments with aflibercept or ranibizumab, the culture without reagents of aflibercept or ranibizumab was adopted as a control culture, and a reagent control was not used; (4) the ARPE-19 cell line was used as the only experimental model of human RPE cells in this study, while other human RPE cells with melanin pigmentations such as primary human RPE cells and induced pluripotent stem cell-derived RPE cells were not adopted; (5) the culture period of ARPE-19 cells in our experiments was relatively short, and melanin pigmentation was not presented in the ARPE-19 cells used; (6) the expression patterns of cytokines secreted by ARPE-19 cells were limitedly evaluated in our in vitro model, and the practical biological effects of the secreted cytokines were not verified by in vivo experiments, e.g., the chick chorioallantoic membrane assay.

In the future, further basic research is required to examine the specific expression patterns of cytokines secreted by native human RPE cells with melanin pigmentations under various experimental conditions with light exposure and anti-VEGF agents. In addition, another complementary experiment to investigate cell surface proteins using mass cytometry [79] is hoped for in order to gain a better understanding of the pathophysiology of chorioretinal diseases related to the RPE.

## 5. Conclusions

This study showed that ARPE-19 cells secrete many inflammatory cytokines simultaneously, and the comprehensive effect of the secreted cytokines is presumed to be an angiogenesis-related function. Continuous irradiation of visible light suppressed most of the cytokines secreted by ARPE-19 cells; however, VEGF-A production was maintained. bFGF expression was increased in a light intensity-dependent manner and also elevated under blue light irradiation compared to green or red light. Anti-VEGF antibodies increased the secretions of inflammatory cytokines including IL-6, IL-8, bFGF and MCP-1, and those elevations may be a complementary response under VEGF suppression.

## Figures and Tables

**Figure 1 biomedicines-09-01333-f001:**
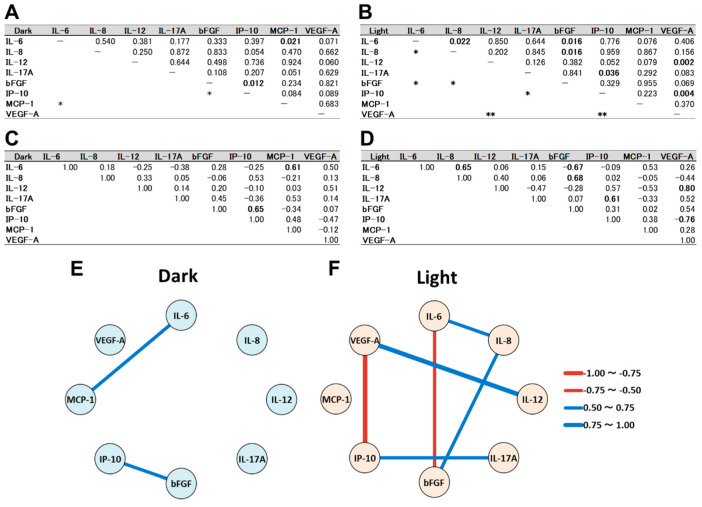
Correlation between inflammatory cytokine levels in the culture supernatant of ARPE-19 cells using Pearson’s correlation analysis. (**A**,**B**) *p* values and (**C**,**D**) Pearson’s correlation coefficients in the correlation analysis between the cytokine levels in (**A**,**C**) dark culture and (**B**,**D**) light-irradiated culture are shown. Undirected graphs indicate significant correlations between the cytokine levels in (**E**) dark culture and (**F**) light-irradiated culture. The red line exhibits a negative correlation, and the blue line means a positive correlation. The thickness of the lines denotes the magnitude of the correlation coefficient. *: *p* < 0.05, **: *p* < 0.01.

**Figure 2 biomedicines-09-01333-f002:**
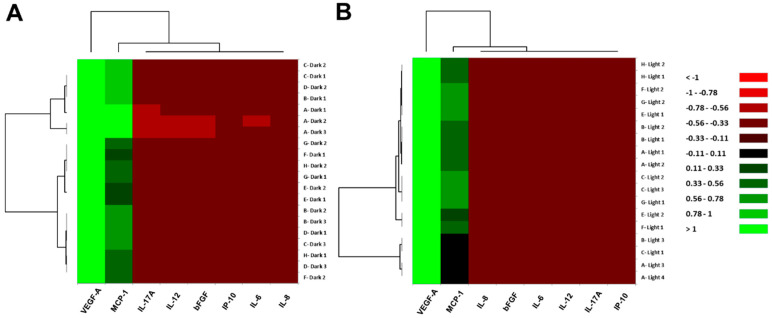
Hierarchical cluster analysis of inflammatory cytokine expression patterns in the culture supernatant of ARPE-19 cells. Heat maps of the cytokines in (**A**) dark culture and (**B**) light-irradiated culture are shown. The vertical axis indicates each experimental culture (uppercase letter alphabet: plate title; numeral: well number), and the horizontal axis denotes individual cytokines. Color scale: red colors mean low values, and black to green colors exhibit middle to high values.

**Figure 3 biomedicines-09-01333-f003:**
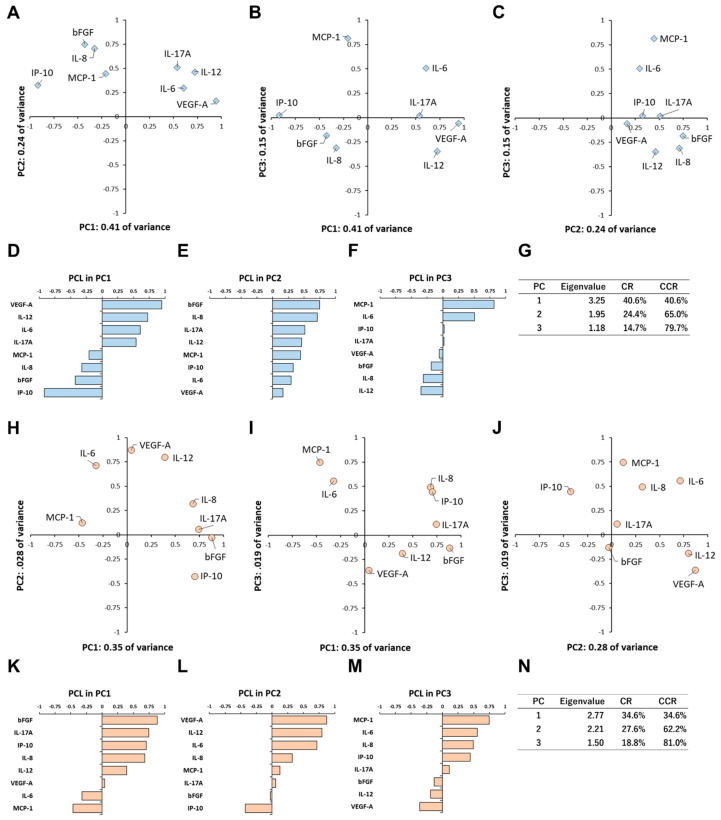
Principal component analysis of inflammatory cytokine expression patterns in the culture supernatant of ARPE-19 cells incubated in (**A**–**G**) dark culture or (**H**–**N**) light-irradiated culture. In dark culture, biplots of PCLs in (**A**) PC1 and PC2, (**B**) PC1 and PC3 and (**C**) PC2 and PC3 are shown. PCLs of (**D**) PC1, (**E**) PC2 and (**F**) PC3 are presented. (**G**) Eigenvalues, CR and CCR of PC1, PC2 and PC3 are described. In the light-irradiated culture, biplots of PCLs in (**H**) PC1 and PC2, (**I**) PC1 and PC3 and (**J**) PC2 and PC3 are shown. PCLs of (**K**) PC1, (**L**) PC2 and (**M**) PC3 are presented. (**N**) Eigenvalues, CR and CCR of PC1, PC2 and PC3 are described. CCR: cumulative contribution ratio, CR: contribution ratio, PC: principal component, PCL: principal component loading, PC1: first principal component, PC2: second principal component, PC3: third principal component.

**Figure 4 biomedicines-09-01333-f004:**
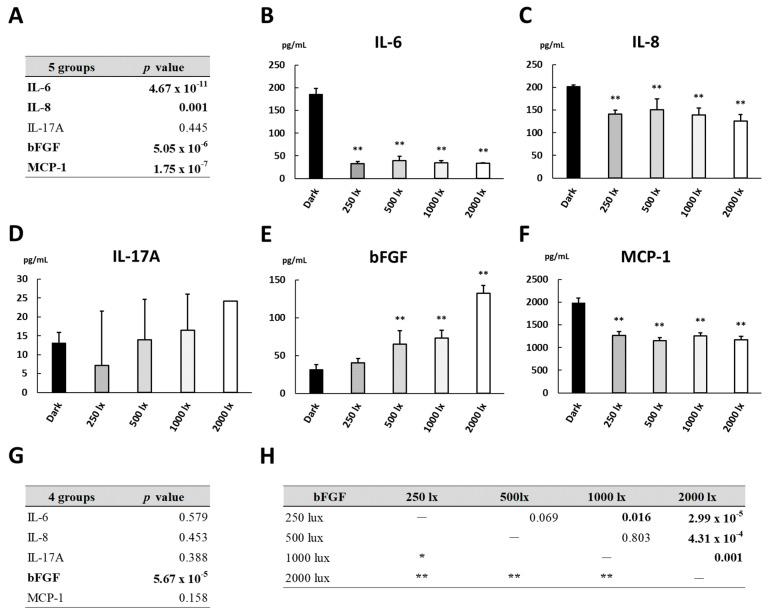
Cytokine levels in the culture supernatant of ARPE-19 cells incubated for 24 h in the dark or under light irradiation with intensities of 250, 500, 1000 and 2000 lux. Five groups consisted of the dark culture and four light-irradiated cultures, and four groups were composed of four light-irradiated cultures. Cytokine levels among (**A**) 5 groups and (**G**) 4 groups were compared by non-repeated measures ANOVA. (**B**–**F**) Dunnett’s test was used for multiple comparisons of cytokine levels between each light-irradiated culture and dark culture as control. (**H**) *p* values in all pairwise comparisons of bFGF levels among light-irradiated cultures at indicated illuminances were obtained by the Tukey–Kramer test. *n* = 4 in each group. lx: lux, *: *p* < 0.05, **: *p* < 0.01.

**Figure 5 biomedicines-09-01333-f005:**
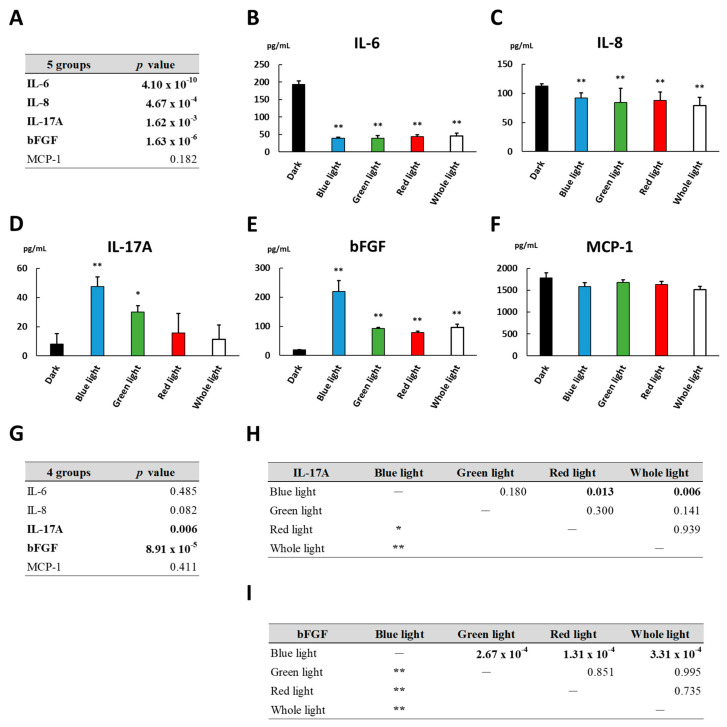
Cytokine levels in the culture supernatant of ARPE-19 cells incubated for 24 h in the dark or under blue, green, red or whole light irradiation. Five groups consisted of the dark culture and four light-irradiated cultures, and four groups were composed of four light-irradiated cultures. Cytokine levels among (**A**) 5 groups and (**G**) 4 groups were compared by non-repeated measures ANOVA. (**B**–**F**) Dunnett’s test was used for multiple comparisons of cytokine levels between each light-irradiated culture and dark culture as control. (**H**,**I**) *p* values in all pairwise comparisons of IL-17A and bFGF levels among light-irradiated cultures at indicated light wavelengths were obtained by the Tukey–Kramer test. *n* = 4 in each group. *: *p* < 0.05, **: *p* < 0.01.

**Figure 6 biomedicines-09-01333-f006:**
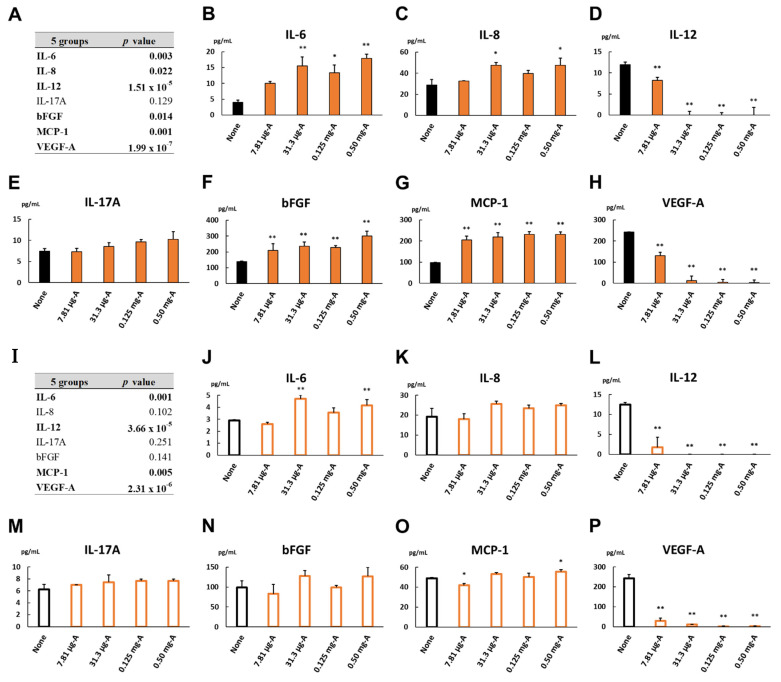
Cytokine levels in the culture supernatant of ARPE-19 cells incubated with 7.81 μg/mL, 31.3 μg/mL, 0.125 mg/mL and 0.50 mg/mL of aflibercept or no concentration as control for 24 h (**A**–**H**) in the dark or (**I**–**P**) under light irradiation. Five groups consisted of four aflibercept-exposed cultures and control. In the dark culture, (**A**) cytokine levels among 5 groups were compared by non-repeated measures ANOVA. (**B**–**H**) Dunnett’s test was used for multiple comparisons of cytokine levels between each aflibercept-exposed culture and control. In the light-irradiated culture, (**I**) cytokine levels among 5 groups were compared. (**J**–**P**) Multiple comparisons between each aflibercept-exposed culture and control were performed. *n* = 3 in each group. *: *p* < 0.05, **: *p* < 0.01.

**Figure 7 biomedicines-09-01333-f007:**
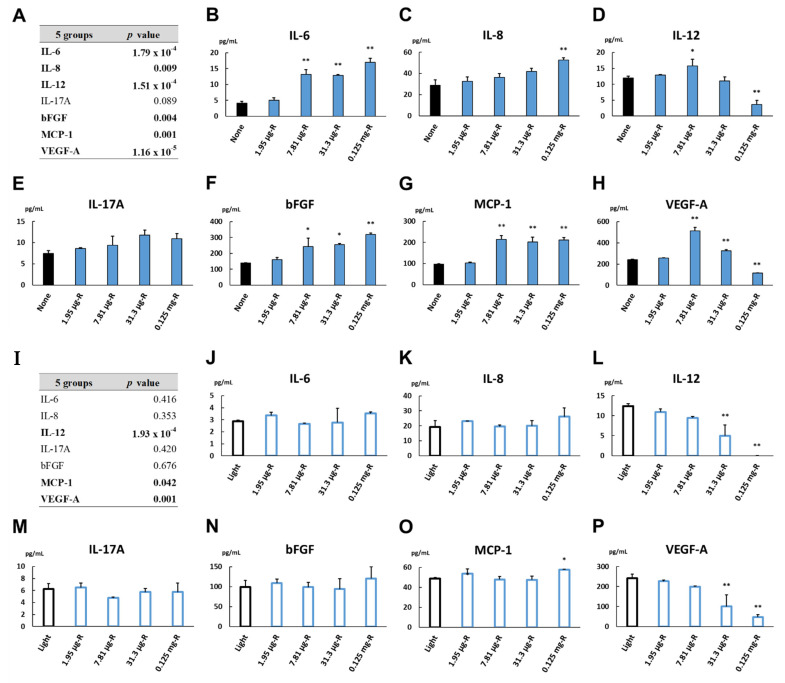
Cytokine levels in the culture supernatant of ARPE-19 cells incubated with 1.95 μg/mL, 7.81 μg/mL, 31.3 μg/mL and 0.125 mg/mL of ranibizumab or no concentration as control for 24 h (**A**–**H**) in the dark or (**I**–**P**) under light irradiation. Five groups consisted of four ranibizumab-exposed cultures and control. In the dark culture, (**A**) cytokine levels among 5 groups were compared by non-repeated measures ANOVA. (**B**–**H**) Dunnett’s test was used for multiple comparisons of cytokine levels between each ranibizumab-exposed culture and control. In the light-irradiated culture, (**I**) cytokine levels among 5 groups were compared. (**J**–**P**) Multiple comparisons between each ranibizumab-exposed culture and control were performed. *n* = 3 in each group. *: *p* < 0.05, **: *p* < 0.01.

**Table 1 biomedicines-09-01333-t001:** Levels of inflammatory cytokines in the culture supernatant of ARPE-19 cells incubated in the dark or under continuous light irradiation of a fluorescent lamp.

Category	Dark		Light		*p* Value	Detection Range		
*n*	20		18					
	Detectable	Level	Detectable	Level				
	Samples (%)	Mean ± SD (Median)	Samples (%)	Mean ± SD (Median)				
**PDGF-BB**	0 (0)	0	5 (27.8)	0.69 ± 1.16 (0)	**5.83 × 10^−3^**	0	to	26,155
IL-1β	0 (0)	0	0 (0)	0	—	0.33	to	6228
IL-1ra	16 (80.0)	8.06 ± 4.69 (9.07)	14 (77.8)	10.5 ± 6.63 (11.5)	0.100	4.97	to	132,198
IL-2	2 (10.0)	0.12 ± 0.36 (0)	4 (22.2)	0.30 ± 0.57 (0)	0.128	0.54	to	18,791
IL-4	1 (5.0)	0.08 ± 0.34 (0)	2 (11.1)	0.15 ± 0.44 (0)	0.279	0.80	to	4272
IL-5	0 (0)	0	0 (0)	0	—	1.61	to	5381
**IL-6**	20 (100)	120.2 ± 63.3 (98.9)	18 (100)	70.0 ± 38.4 (56.6)	**0.003**	1.39	to	21,699
**IL-7**	20 (100)	3.89 ± 0.42 (3.89)	18 (100)	3.39 ± 0.71 (3.32)	**0.005**	0.49	to	11,208
**IL-8**	20 (100)	151.3 ± 39.9 (140.5)	18 (100)	112.1 ± 23.1 (110.3)	**4.18 × 10^−4^**	1.38	to	25,457
**IL-9**	19 (95.0)	4.18 ± 1.64 (4.12)	18 (100)	8.17 ± 2.35 (8.89)	**2.31 × 10^−75^**	1.35	to	9686
IL-10	16 (80.0)	2.47 ± 1.32 (2.88)	17 (94.4)	2.78 ± 0.82 (2.93)	0.196	2.17	to	34,225
IL-12	20 (100)	47.2 ± 5.57 (47.0)	18 (100)	46.6 ± 3.59 (46.3)	0.351	2.33	to	35,172
**IL-13**	7 (35.0)	0.84 ± 1.19 (0)	2 (11.1)	0.25 ± 0.72 (0)	**0.038**	0.32	to	8051
IL-15	20 (100)	3.68 ± 1.27 (3.78)	17 (94.4)	3.95 ± 1.18 (4.14)	0.250	1.50	to	6322
**IL-17A**	16 (80.0)	11.3 ± 6.51 (13.8)	17 (94.4)	19.6 ± 10.0 (23.0)	**0.002**	1.82	to	32,080
Eotaxin	3 (15.0)	1.54 ± 3.86 (0)	0 (0)	0	0.050	0	to	25,602
**bFGF**	20 (100)	30.7 ± 10.4 (29.6)	18 (100)	116.4 ± 31.5 (130.6)	**6.50 × 10^−145^**	2.90	to	3719
G-CSF	19 (95.0)	3.17 ± 1.05 (3.49)	15 (83.3)	3.49 ± 1.52 (3.92)	0.227	1.67	to	8005
GM-CSF	0 (0)	0	0 (0)	0	—	2.62	to	13,998
IFN-γ	7 (35.0)	3.56 ± 5.31 (0)	4 (22.2)	1.47 ± 2.85 (0)	0.073	4.57	to	23,703
IP-10	19 (95.0)	24.4 ± 17.9 (18.3)	15 (83.3)	25.8 ± 17.9 (20.8)	0.404	11.0	to	37,606
**MCP-1**	20 (100)	1652.5 ± 392.0 (1688.1)	18 (100)	1229.9 ± 396.3 (1208.7)	**0.001**	1.52	to	19,447
MIP-1α	6 (30.0)	0.39 ± 0.62 (0)	2 (11.1)	0.14 ± 0.40 (0)	0.077	0.05	to	1083
MIP-1β	1 (5.0)	0.05 ± 0.23 (0)	1 (5.6)	0.07 ± 0.30 (0)	0.418	0.29	to	3679
RANTES	0 (0)	0	1 (5.6)	0.14 ± 0.59 (0)	0.149	1.83	to	5365
TNFα	0 (0)	0	0 (0)	0	—	3.26	to	20,533
VEGF-A	20 (100)	3963.8 ± 726.1 (4102.0)	18 (100)	3896.7 ± 553.1 (4043.5)	0.376	2.42	to	38,721

Cytokine levels are presented in units of pg/mL. Cytokines with significant differences are shown in bold. *p* value: cytokine level in dark culture vs. light-irradiated culture.

**Table 2 biomedicines-09-01333-t002:** Levels of inflammatory cytokines in culture supernatant of ARPE-19 cells incubated with anti-VEGF antibody in the dark.

Category	Dark			*p* Value		
Anti-VEGF Antibody	None	Aflibercept	Ranibizumab	None		Aflibercept
	Level	Level	Level	vs.		vs.
	Mean ± SD	Mean ± SD	Mean ± SD	Aflibercept	Ranibizumab	Ranibizumab
PDGF-BB	1.90 ± 0.54	1.51 ± 0	2.64 ± 2.31	0.211	0.351	0.281
IL-1β	0	0.66 ± 0.11	0.70 ± 0.06	**0.007**	**0.002**	0.349
IL-1ra	10.5 ± 0.62	12.2 ± 0.60	11.7 ± 0	0.053	**0.049**	0.211
IL-2	0.51 ± 0.72	0.39 ± 0.55	0	0.434	0.211	0.211
IL-4	0	0	0	—	—	—
IL-5	0	0	0	—	—	—
**IL-6**	4.08 ± 0.59	17.9 ± 1.20	17.0 ± 1.31	**0.002**	**0.003**	0.263
IL-7	2.91 ± 0.56	2.20 ± 0.15	3.50 ± 0.28	0.112	0.156	**0.014**
**IL-8**	28.7 ± 5.37	47.4 ± 6.80	52.8 ± 1.99	**0.046**	**0.014**	0.200
IL-9	2.29 ± 0.09	3.25 ± 0	3.88 ± 0.70	**0.002**	**0.043**	0.167
IL-10	0	0	0	—	—	—
**IL-12**	11.9 ± 1.32	0	3.68 ± 0.30	**0.003**	**0.007**	**0.002**
IL-13	0.94 ± 0.03	0.41 ± 0	0.54 ± 0.06	**0.001**	**0.008**	0.054
IL-15	2.53 ± 0.02	5.51 ± 0.46	5.22 ± 0.06	**0.006**	**1.26 × 10^−4^**	0.238
**IL-17A**	7.46 ± 0.62	10.3 ± 1.83	10.9 ± 1.22	0.088	**0.035**	0.359
Eotaxin	0	0	0	—	—	—
**bFGF**	139.0 ± 2.69	301.3 ± 29.8	319.3 ± 8.35	**0.008**	**0.001**	0.249
G-CSF	0	0	0	—	—	—
GM-CSF	0	0	0	—	—	—
IFN-γ	0	0	0	—	—	—
IP-10	0	0	0	—	—	—
**MCP-1**	97.7 ± 1.00	231.1 ± 11.8	211.7 ± 11.0	**0.002**	**0.002**	0.116
MIP-1α	0	0.13 ± 0.18	0	0.211	—	0.211
MIP-1β	0	0	0	—	—	—
RANTES	0	0	0	—	—	—
TNFα	0	0	0	—	—	—
**VEGF-A**	243.0 ± 2.51	3.24 ± 0.28	115.2 ± 3.04	**2.77 × 10^−55^**	**2.38 × 10^−4^**	**1.86 × 10^−4^**

ARPE-19 cells were cultured with 0.50 mg/mL aflibercept, 0.125 mg/mL ranibizumab or neither (no reagent) as control in the dark for 24 h. Levels of IL-6, IL-8, IL-12, L-17A, bFGF, MCP-1 and VEGF-A were higher than 10 pg/mL in at least one of the three cultures. *n* = 4 in each group.

**Table 3 biomedicines-09-01333-t003:** Levels of inflammatory cytokines in the culture supernatant of ARPE-19 cells incubated with anti-VEGF antibody under light irradiation.

Category	Light			*p* value		
Anti-VEGF Antibody	None	Aflibercept	Ranibizumab	None		Aflibercept
	Level	Level	Level	vs.		vs.
	Mean ± SD	Mean ± SD	Mean ± SD	Aflibercept	Ranibizumab	Ranibizumab
PDGF-BB	2.11 ± 0.73	2.02 ± 0	1.01 ± 1.43	0.442	0.162	0.211
IL-1β	0	0	0	—	—	—
IL-1ra	11.2 ± 1.96	15.9 ± 1.15	14.0 ± 2.04	**0.029**	0.106	0.190
IL-2	0.75 ± 0.13	1.34 ± 0.17	0.90 ± 0.38	**0.011**	0.280	0.137
IL-4	0	0	0	—	—	—
IL-5	0	0	0	—	—	—
IL-6	2.88 ± 0.09	4.16 ± 0.47	3.53 ± 0.12	**0.008**	**0.003**	0.102
IL-7	3.61 ± 1.13	0.74 ± 0	1.66 ± 0.62	**0.021**	0.060	0.085
IL-8	19.2 ± 4.30	24.9 ± 0.88	26.1 ± 5.90	0.087	0.109	0.401
IL-9	2.00 ± 0.08	1.66 ± 0.43	0.71 ± 1.00	0.117	**0.047**	0.173
IL-10	0	0	0	—	—	—
**IL-12**	12.4 ± 0.55	0	0	**4.02 × 10^−55^**	**4.02 × 10^−55^**	—
IL-13	1.18 ± 0.11	0.21 ± 0.29	0.21 ± 0.29	**0.006**	**0.006**	0.550
IL-15	3.41 ± 0.23	4.06 ± 0.45	3.74 ± 0.18	0.058	0.098	0.226
IL-17A	6.23 ± 0.87	7.67 ± 0.31	5.73 ± 1.53	0.061	0.330	0.110
Eotaxin	0	0	0	—	—	—
bFGF	99.2 ± 16.3	127.0 ± 22.5	120.4 ± 29.3	0.100	0.180	0.412
G-CSF	0	0	0	—	—	—
GM-CSF	0	0	0	—	—	—
IFN-γ	0	0	0	—	—	—
IP-10	0	0	0	—	—	—
**MCP-1**	48.8 ± 0.99	55.6 ± 2.19	57.7 ± 0.26	**0.008**	**0.001**	0.154
MIP-1α	0.26 ± 0	0.13 ± 0.18	0.34 ± 0.11	0.136	0.136	0.151
MIP-1β	0.19 ± 0.17	0	0	0.110	0.110	—
RANTES	0	0	0	—	—	—
TNFα	0	0	0	—	—	—
**VEGF-A**	242.2 ± 20.1	3.15 ± 0.42	47.4 ± 12.4	**2.67 × 10^−4^**	**0.001**	**0.019**

ARPE-19 cells were cultured with 0.50 mg/mL aflibercept, 0.125 mg/mL ranibizumab or neither under light irradiation for 24 h. Levels of IL-12, MCP-1 and VEGF-A were more than 10 pg/mL in at least one of the three cultures. *n* = 4 in each group.

**Table 4 biomedicines-09-01333-t004:** Cytokine levels in the culture supernatant of ARPE-19 cells incubated with anti-VEGF antibody: comparison between dark and light irradiation.

Anti-VEGF Antibody	Aflibercept			Ranibizumab		
Category	Dark	Light	*p* Value	Dark	Light	*p* Value
	Level	Level	Dark vs. Light	Level	Level	Dark vs. Light
	Mean ± SD	Mean ± SD		Mean ± SD	Mean ± SD	
PDGF-BB	1.51 ± 0	2.02 ± 0	1.000	2.64 ± 2.31	1.01 ± 1.43	0.243
IL-1β	0.66 ± 0.11	0	**0.007**	0.70 ± 0.06	0	**0.002**
**IL-1ra**	12.2 ± 0.60	15.9 ± 1.15	**0.028**	11.7 ± 0	14.0 ± 2.04	0.126
IL-2	0.39 ± 0.55	1.34 ± 0.17	0.073	0	0.90 ± 0.38	**0.040**
IL-4	0	0	—	0	0	—
IL-5	0	0	—	0	0	—
**IL-6**	17.9 ± 1.20	4.16 ± 0.47	**0.002**	17.0 ± 1.31	3.53 ± 0.12	**0.002**
IL-7	2.20 ± 0.15	0.74 ± 0	**0.003**	3.50 ± 0.28	1.66 ± 0.62	**0.031**
**IL-8**	47.4 ± 6.80	24.9 ± 0.88	**0.022**	52.8 ± 1.99	26.1 ± 5.90	**0.013**
IL-9	3.25 ± 0	1.66 ± 0.43	**0.017**	3.88 ± 0.70	0.71 ± 1.00	**0.034**
IL-10	0	0	—	0	0	—
IL-12	0	0	—	3.68 ± 0.30	0	**0.002**
IL-13	0.41 ± 0	0.21 ± 0.29	0.211	0.54 ± 0.06	0.21 ± 0.29	0.128
IL-15	5.51 ± 0.46	4.06 ± 0.45	**0.043**	5.22 ± 0.06	3.74 ± 0.18	**0.004**
**IL-17A**	10.3 ± 1.83	7.67 ± 0.31	0.094	10.9 ± 1.22	5.73 ± 1.53	**0.032**
Eotaxin	0	0	—	0	0	—
**bFGF**	301.3 ± 29.8	127.0 ± 22.5	**0.011**	319.3 ± 8.35	120.4 ± 29.3	**0.006**
G-CSF	0	0	—	0	0	—
GM-CSF	0	0	—	0	0	—
IFN-γ	0	0	—	0	0	—
IP-10	0	0	—	0	0	—
**MCP-1**	231.1 ± 11.8	55.6 ± 2.19	**0.001**	211.7 ± 11.0	57.7 ± 0.26	**0.001**
MIP-1α	0.13 ± 0.18	0.13 ± 0.18	0.550	0	0.34 ± 0.11	**0.026**
MIP-1β	0	0	—	0	0	—
RANTES	0	0	—	0	0	—
TNFα	0	0	—	0	0	—
**VEGF-A**	3.24 ± 0.28	3.15 ± 0.42	0.407	115.2 ± 3.04	47.4 ± 12.4	**0.009**

Concentrations of aflibercept and ranibizumab were 0.50 mg/mL and 0.125 mg/mL, respectively. Levels of IL-1ra, IL-6, IL-8, IL-17A, bFGF, MCP-1 and VEGF-A in dark culture with aflibercept or ranibizumab were more than 10 pg/mL.

## Data Availability

The data used to support the findings of this study are available from the corresponding author upon request.

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
