# Peer review of "Profiles of Cytokines Secreted by ARPE-19 Cells Exposed to Light and Incubated with Anti-VEGF Antibody"

_biomedicines, 2021, doi:10.3390/biomedicines9101333_

Round 1

Reviewer 1 Report

Any comments

Author Response

Responses to the Comments of Reviewers

We thank the editor and referees for taking their time to review our manuscript. Following Reviewer #1’s suggestions, we have made some corrections and clarifications in the manuscript. All the coauthors have read and agreed with the changes made in the revised manuscript.

We hope that the corrections and revisions are satisfactory, and the revised version will be acceptable for publication. We thank once again Reviewer #1 for the constructive comments. Our responses to the comments and the changes are summarized below.

Responses to Reviewer #1

Comment. English language and style are minor spell check required.

Reply:

We thank the reviewer for the comment. We proofread the writing in the manuscript, and corrected it carefully.

Reviewer 2 Report

This is mainly analytical manuscript with no mechanistic studies and that is why the authors’ conclusions on the relationship between the obtained results and angiogenesis are very limited. The authors use only one angiogenesis marker VEGF. On the other side the use of anit-VEGF antibodies in the conditions of the experiments is of a clinical value. The main disadvantage of that work is the use of ARPE-19 cell line as the only experimental model. Although still many papers appear with these cells they should not be treated as the only biological model. RPE cells obtained from the iPSCs are mostly recommended. The authors should thoroughly discuss the limitations following from the usage of these cells. Please refer e.g. to papers of Kozlowski and Blasiak et al. showing that ARPE-19 cells in culture are mostly senescent and consequently release several factors not present in the normal retina. Authors claim that they attempted to mimic natural RPE exposure by an additional source of light. Is that correct? What about the applied modified cell media related to the natural RPE environment? Anyway, in my opinion the authors investigated a combined exposure to light and chemicals, in which exposure to light was rather additional than natural. However, the comparison of the action of blue, green and red irradiations is valuable.

Title: is too long and somehow confusing; consider the following:

  1. Using the RPE and VEGF abbreviations instead of full names
  2. Exposed to Visible-Light Irradiation -> exposed to light
  3. and Anti-Vascular Endothelial Growth Factor Antibody -> and incubated with anti-VEGF

Abstract: Which anti-VEGF antibody was used? Did the authors full spectrum of visible light or blue light? Comprehensive effect? This is unclear and must be clarified. The last sentence is justified neither by results, nor methodology. No words about angiogenesis research (methodology).

Introduction: Introduction consist of several paragraphs, each is associated with the main subject of the manuscript, but they are principally independent of each other’s. A mor smooth transition between them should be applied, e.g., the last fragment of the preceding paragraph should relate to the subsequent. Sentence: “Various…” is not completely clear and likely grammatically improper. What do the authors mean using the word “monolayer”? This word is applied to monolayer of RPE cells as well as a monolayer composed of RPE, Bruch’s membrane and choriocapillaris.

Methods are well written, but why two different multiple comparison tests were used?

Results are well presented and discussion lacks critics on the use of ARPE-19 cells as the only experimental model (see above).

Author Response

Responses to the Comments of Reviewers

We thank the editor and referees for taking their time to review our manuscript. Following Reviewer #2’s suggestions, we have made some corrections and clarifications in the manuscript. All the coauthors have read and agreed with the changes made in the revised manuscript.

We hope that the corrections and revisions are satisfactory, and the revised version will be acceptable for publication. We thank once again Reviewer #2 for the constructive comments. Our responses to the comments and the changes are summarized below.

Responses to Reviewer #2

Comment 1.

Title: is too long and somehow confusing; consider the following:

1: Using the RPE and VEGF abbreviations instead of full names

2: Exposed to Visible-Light Irradiation -> exposed to light

3: and Anti-Vascular Endothelial Growth Factor Antibody -> and incubated with anti-VEGF

Reply 1:

We thank the reviewer for the comment. Based on the previous comments, we hope that the readers can understand our experimental results with accuracy. So, we changed the words “Human Retinal Pigment Epithelial Cells” to “ARPE-19 Cells” and corrected the title as follows: “Profiles of Cytokines Secreted by ARPE-19 Cells Exposed to Light and Incubated with Anti-VEGF Antibody” (lines 3-4).

Comment 2.

Abstract: Which anti-VEGF antibody was used? Did the authors full spectrum of visible light or blue light? Comprehensive effect? This is unclear and must be clarified. The last sentence is justified neither by results, nor methodology. No words about angiogenesis research (methodology).

Reply 2:

We thank the reviewer for the comment. As you pointed out, the description of the previous abstract is inappropriate because some contents derived from our experimental results may include uncertain interpretations. Therefore, we reviewed the previous abstract, and corrected it appropriately based on the reviewer’s comment, in which the word “angiogenesis” is not included. (lines 33-50, abstract).

Comment 3.

Introduction: Introduction consist of several paragraphs, each is associated with the main subject of the manuscript, but they are principally independent of each other’s. A mor smooth transition between them should be applied, e.g., the last fragment of the preceding paragraph should relate to the subsequent. Sentence: “Various…” is not completely clear and likely grammatically improper. What do the authors mean using the word “monolayer”? This word is applied to monolayer of RPE cells as well as a monolayer composed of RPE, Bruch’s membrane and choriocapillaris.

Reply 3:

We thank the reviewer for the comment. We reviewed overall structure of the introduction section, and corrected the sentences to be easy for reading it smoothly. (lines 51-96, Introduction). In introduction section, we described the word “monolayer” in the citation (lines 52-54) according to the reference document, and deleted the word “monolayer” in the citation (lines 56) as follows: “the outer BRB consists of the RPE, Bruch’s membrane and choriocapillaris”. We also change the word “various” to “some” (lines 56).

Comment 4.

Methods are well written, but why two different multiple comparison tests were used?

Reply 4:

We thank the reviewer for the comment. As you pointed out, the sentence “~ was used for parametric comparisons of multiple groups” is duplicate description, and confuses the readers about the differences of statistical interpretations between non-repeated measures ANOVA and Tukey-Kramer test. In parametric comparisons among multiple groups, non-repeated measures ANOVA is prerequisite statistical equation. In the case that Tukey-Kramer test or Dunnett’s test indicates some significant difference, but non-repeated measures ANOVA denotes no significant difference, it is judged that there is no significant difference among multiple groups. We corrected the sentences as follows: “Dunnett’s test followed by post hoc non-repeated measures ANOVA was used for parametric comparisons with control, and Tukey-Kramer test was adopted for parametric all pairwise comparisons. (lines 170-172).

Comment 5.

Results are well presented and discussion lacks critics on the use of ARPE-19 cells as the only experimental model (see above).

Reply 5:

We thank the reviewer for the comment. As you pointed out, our research has several limitations in human RPE cell type, cell maturity, expression of melanin pigmentation and reagent control. The main purpose of our study is to examine particular expression patterns of cytokines secreted by human RPE cells under continuous irradiation of visible light. As for the light exposure experiments, we previously tried to use a human fetal primary RPE cell obtained from LONZA (Walkersville, MD, USA, Cat#: 00194987), and evaluated the expression patterns of cytokines secreted by the primary RPE cells. The primary RPE cells expressed melanin pigmentation in short-term culture (approximately 1 week), however, the expressions of melanin pigmentation were not uniform. ARPE-19 cells and the primary RPE cells are thickly overlayed in whole or part in long-term culture (over a few weeks or more). We consider that individual cells must be equally exposed to the light, and monolayer of human RPE cells is necessary to obtain reproducible results in the light exposure experiment. Furthermore, the expression of melanin pigmentation caused excessive photooxidative stress in the human RPE cells incubated under the light irradiation. In our preliminary experiments, cell death was not induced until 12 hours, but occurs with apoptosis in the human RPE cells under the light irradiation. On the other hand, the difference in the indicated culture period (24 hours: ARPE-19 cells vs. 12 hours: the human RPE cells) is a significant variation factor in cytokine expressions. Therefore, to avoid confusion and misunderstanding for the results in the expression patterns of secreted cytokines, the experimental results obtained from ARPE-19 cells are only described.

 In this study, the comprehensive effects of secreted cytokines are evaluated by PCA as a multivariate analysis. We carefully interpretate the results of PCA according to principal effect of individual cytokine and its PCL, and hypothesize that the comprehensive effect is presumed to be angiogenesis-related function. However, the hypothesis is a limitedly evaluation from a statistical point of view, and may be uncertain as an evaluation for biological responses. Therefore, we reviewed the manuscript, and deleted the results of PCA in abstract. Furthermore, the hypothesis obtained from PCA results was described minimally in discussion.

As for light exposure condition, ambient light could be defined variously by experimental conditions of light exposure, and is uniform. Continuous light exposure from the fluorescent lamp may not be an ideal but a relatively appropriate condition of light exposure. We previously discussed the light exposure condition in this study as follows:” The color temperature of fluorescent lamp light and the intensity of illumination on the cells were considerable factors in this study. The International Commission on Illumination (the CIE) has introduced well-known standard illuminants in the categories of illuminant A to F [33]. The CIE Standard illuminant D65 in illuminant D represents natural daylight (6504 K) and is one of the standard illuminations used in color science and engineering. The illuminant series F represents various types of fluorescent lighting (6500 K) emitting full-spectrum light similarly with illuminant D65. Regarding intensity of illumination, the intensities of fluorescent lamp light between 500 lx and 15,000 lx were adopted as intensities of ambient light in vivo [34–36]. We also adopted 2000 lx as an intensity of ambient light in vitro [19, 29]. Based on previous reports, we applied 2000 lx on the cells under a fluorescent lamp belonging to the illuminant series F as an experimental condition imitating irradiation of ambient light.” (Reference 24)

Regarding colorless medium for light exposure experiment, the PBS-based medium is adopted in our study. In preliminary experiments, phenol red free DMEM/F12 with 10% FBS was initially used in the light irradiated culture. However, ARPE-19 cells were extinct in the light irradiated culture. The extinction of the cells was also occurred in phenol red free DMEM/F12 with 1% FBS. Based on the results of the preliminary experiments, we consider that some compositions of DMEM/F12 were changed to be cytotoxic substances by the light exposure. Therefore, we created a colorless PBS-based medium, and used it in previous basic researches (Reference 22, 23, 24)

 In discussion section, we described the replies for the reviewer’s comments as follows: (1) the comprehensive effects of secreted cytokines in PCA. In result section: (lines 248-250), (lines 252-253), (lines 255-256), (lines 264-266), (lines 268-270), (lines 272-274). In discussion section: (lines 416-417), (lines 438-440), (lines 568-569). (2) the usage of ARPE-19 cell line (lines 534-543), (lines 551-554). (3) reagent control in anti-VEGF exposure experiment (lines 510-517), (lines 548-551). (4) Melanin pigmentation in light exposure experiment (lines 518-533), (lines 554-555). (5) cell senescence in ARPE-19 cells (lines 534-543).

Reviewer 3 Report

In this manuscript, the author analyzed cytokines secretion in PRE cell under different light conditions. I really appreciate the work. However, it would be better to have an extra paragraph to summarize the research results (eg: conclusion) and also indicating the values of these results to the future disease treatment/drug development.  It would be also helpful for readers to further understand the research after reading so many figures. 

Author Response

Responses to the Comments of Reviewers

We thank the editor and referees for taking their time to review our manuscript. Following Reviewer #3’s suggestions, we have made some corrections and clarifications in the manuscript. All the coauthors have read and agreed with the changes made in the revised manuscript.

We hope that the corrections and revisions are satisfactory, and the revised version will be acceptable for publication. We thank once again Reviewer #3 for the constructive comments. Our responses to the comments and the changes are summarized below.

Responses to Reviewer #3

Comment. In this manuscript, the author analyzed cytokines secretion in PRE cell under different light conditions. I really appreciate the work. However, it would be better to have an extra paragraph to summarize the research results (eg: conclusion) and also indicating the values of these results to the future disease treatment/drug development. 

Reply:

We thank the reviewer for the comment. We added extra paragraphs summarizing the results of our research as follows: (lines 223-227, in results), (lines 236-238, in results), (lines 248-250, in results), (lines 252-253, in results), (lines 255-256, in results), (lines 264-266, in results), (lines 268-270, in results), (lines 272-274, in results), (lines 287-290, in results), (lines 303-307, in results), (lines 319-321, in results). (lines 332-333, in results), (lines 349-351, in results), (lines 362-366, in results), (lines 377-380, in results), (lines 392-394, in results), (lines 405-408, in results).

Round 2

Reviewer 2 Report

Title and further when necessary – please consider specifying VEGF-A

Abstract, the first sentence: “…in cytokines…” – please add “in the eye” or “in the retina”

Introduction: “AMD is a leading cause of blindness among the elderly in developed countries,…” – “the” would be better than “a”

3.8. Levels of Inflammatory Cytokines with Anti-VDGF Antibody – please correct VDGF

Section 3.10 – the original title is better for me than the changed one

3.12. Cytokine Levels under Light Irradiation Exposed to Aflibercept at Various Concentrations – please change to e.g., Cytokine Levels in ARPE-19 Cells Exposed to Light and Incubated with Light

please adjust the titles of proceedings and subsequent sections

I am afraid that most of figures may not be readable “in print”

  1. Conclusions – please improve the style, as in the present form it is intricate

Author Response

Responses to the Comments of Reviewers

We thank the editor and referees for taking their time to review our manuscript. Following Reviewer #2’s suggestions, we have made some corrections and clarifications in the manuscript. All the coauthors have read and agreed with the changes made in the revised manuscript.

We hope that the corrections and revisions are satisfactory, and the revised version will be acceptable for publication. We thank once again Reviewer #2 for the constructive comments. Our responses to the comments and the changes are summarized below.

Responses to Reviewer #2

Comment 1.

Title and further when necessary – please consider specifying VEGF-A

Reply 1:

We thank the reviewer for the comment. We also consider that the specify of biological effects in anti-VEGF antibody is an important in this study. As for the anti-VEGF antibody used, Aflibercept binds VEGF-A, -B and PlGF, on the other hand ranibizumab binds only VEGF-A. (lines 521-526). In the common perception, aflibercept and ranibizumab are recognized as one of anti-VEGF antibody, although the specifies of biological effects between aflibercept and ranibizumab are different. Based on the common perception and character limitation of title section, we consider that “Anti-VEGF Antibody” in title (lines 4) is better than “Anti-VEGF-A Antibody”, because “Anti-VEGF Antibody” includes both aflibercept and ranibizumab, but “Anti-VEGF-A Antibody” is applied only to ranibizumab.

On the other hand, the measurement item “VEGF” in Bio-Plex Human Cytokine 27-Plex panel (Bio-Rad, Hercules, CA, USA) is “VEGF-A”, to be exact. Therefore, we correct “VEGF” to “VEGF-A” in the sentences of measurement results (lines 42-43), (lines 48), (lines 202), (lines 206), (lines 210), (lines 233), (lines 247), (lines 267), (lines 281), (lines 314), (lines 319), (lines 329), (lines 331), (lines 332), (lines 341), (lines 346), (lines 349), (lines 361), (lines 377), (lines 392), (lines 405), (lines 433), (lines 437), (lines 442), (lines 448-449), (lines 453), (lines 454), (lines 509), (lines 586), (lines 442), (lines 448-449), Tables, Figures and Supplementary Tables.

“VEGF suppression” (lines 49), (lines 590) means the specify of biological effects in both aflibercept and ranibizumab.

Comment 2.

Abstract: Abstract, the first sentence: “…in cytokines…” – please add “in the eye” or “in the retina”

Reply 2:

We thank the reviewer for the comment. We corrected the sentence as follows: “~is the major source of cytokines in the retina” (lines 34, abstract).

Comment 3.

Introduction: “AMD is a leading cause of blindness among the elderly in developed countries,…” – “the” would be better than “a”

Reply 3:

We thank the reviewer for the comment. We corrected the sentence as follows: “AMD is the leading cause of blindness among the elderly in developed countries,” (lines 63, Introduction).

Comment 4.

3.8. Levels of Inflammatory Cytokines with Anti-VDGF Antibody – please correct VDGF

Reply 4:

We thank the reviewer for the comment. We corrected the subtitle as follows: “3.8. Levels of Inflammatory Cytokines with Anti-VEGF Antibody” (lines 310, Results).

Comment 5.

Section 3.10 – the original title is better for me than the changed one

Reply 5:

We thank the reviewer for the comment. We corrected the subtitle to the original title as follows: “3.10. Cytokine Levels in Aflibercept- or Ranibizumab-exposed ARPE-19 Cells Cultured in Dark and under Light Irradiation” (lines 336-337, Results).

Comment 6.

3.12. Cytokine Levels under Light Irradiation Exposed to Aflibercept at Various Concentrations – please change to e.g., Cytokine Levels in ARPE-19 Cells Exposed to Light and Incubated with Light

 please adjust the titles of proceedings and subsequent sections

Reply 6:

We thank the reviewer for the comment. We consider the regularity of combination words “cells exposed to light”, and “cells incubated with anti-VEGF antibody” according to the title, and corrected the subtitle as follows: “3.12. Cytokine Levels in Light Irradiated Culture Incubated with Aflibercept at Various Concentrations” (lines 369-370, Results).

We also reviewed the regularity of the combination words and corrected the sentences in subsequent sections.

Comment 7.

I am afraid that most of figures may not be readable “in print”

Reply 7:

We thank the reviewer for the comment. We also think that the letters in Tables and Figures are relatively small, and it is not easy for readers to understand our results. We reduced the gap in Figure 1, Figure 2, Figure 3, Figure 6, Figure 7, and enlarged the character in Table 1, Table 2, Table 3, Table 4, and Figure 1, Figure 2, Figure 3, Figure 6, Figure 7, and Table S1, Table S2, Table S3, Table S3.

Comment 8.

5 Conclusions – please improve the style, as in the present form it is intricate

Reply 8:

We thank the reviewer for the comment. We reviewed the conclusions section, and simplified the description of the sentences. (lines 583-590, Conclusions).

Reviewer 3 Report

I don't have extra comment. 

Author Response

Responses to the Comments of Reviewers

We thank the editor and referees for taking their time to review our manuscript. Following Reviewer #3’s suggestions, we have made some corrections and clarifications in the manuscript. All the coauthors have read and agreed with the changes made in the revised manuscript.

We hope that the corrections and revisions are satisfactory, and the revised version will be acceptable for publication. We thank once again Reviewer #3 for the constructive comments. Our responses to the comments and the changes are summarized below.

Responses to Reviewer #3

Comment.

I don't have extra comment.

Reply:

Thank you very much for your valuable time in the peer review for our manuscript.

This manuscript is a resubmission of an earlier submission. The following is a list of the peer review reports and author responses from that submission.

Round 1

Reviewer 1 Report

This study examines the effect of light stress (variable lux and colour, 24 hours), and therapeutic agents that contain anti-VEGF antibodies ().

The Introduction provides some rationale for the study although why combine short term light (24 hours) and VEGF-related therapeutics is not clear.

2. Methods and Research Plan: for the therapeutic agents (aflibercept and  used, no mention is made of the vehicle(s) for these reagents. For the experiments, it is critical to use the vehicles as controls. The two agents are also different in mode of action against VEGF, and this has not been discussed in the paper (either Methods or Discussion). Aflibercept is a recombinant fusion protein that  mimic of VEGFR-1 and -2 and the Fc region of a human immunoglobulin G1 (IgG1), and so binds VEGF-A, -B and placental growth factor with very high affinity. In contrast, ranibizumab is comprised of a humanised monoclonal antibody fragment that binds VEGF-A with high affinity (stops VEGF-A binding receptors, VEGFR-1 and VEGFR-2).

The ARPE19 cell line is frequently used as a model but including adult primary human RPE cultures is important; this is noted as a limitation by the authors. Complete details on how many repeats for each experiment are also required, and the details are not always easy to follow in places.

3. Results

The initial analysis of the cytokine levels using a principal component approach is confusing and why the detection rates of 80% (of what?), and levels of more than 10pg/ml are used is not well explained. Am also unclear if these criteria were applied for the aflibercept and ranibizumab experiments. The correlations between the selected cytokines and growth factors at best, shows association.

4. Discussion

The Discussion is very long and not always easy to follow. The secreted cytokine and growth factor profile of ARPE19 cells in this study (IL6, -8, -12, -17A, bFGF, IP-10, MCP! and VEGF) is reported, with the conclusion that the primary effect of these is angiogenesis - there are no experiments conducted to support this (although much in the literature). The combination of light and the anti-VEGF drugs requires further justification.

The authors also discuss melanin in RPE cells (p.16) however ARPE19 cells have very low levels of dark melanin, and the relevance is uncertain.

The complex interplay of cytokines and bFGF are interesting but further investigation is required to identify the modes of action, including for example, expression patterns of receptors for these factors in th retina, RPE and choroid.

Figures

The Figures showing effects of light  exposure and then dark should be combined in the same figure (same graphs, filled and empty bars) to allow easier comparison between conditions. This also applies to the graphs showing concentration effects for the two drugs used under light and dark conditions.

Author Response

Responses to the Comments of Reviewers

 We thank the editor and referees for taking their time to review our manuscript. Following Reviewer #1’s suggestions, we have made some corrections and clarifications in the manuscript. All the coauthors have read and agreed with the changes made in the revised manuscript.

 We hope the corrections and revisions are satisfactory, and the revised version will be acceptable for publication. We thank once again Reviewer #1 for the constructive comments. Our responses to the comments and the changes are summarized below.

Responses to Reviewer #1

Comment 1. The Introduction provides some rationale for the study although why combine short term light (24 hours) and VEGF-related therapeutics is not clear.

 We thank the reviewer for the comment. We added the appropriate sentence as follows: “At present, anti-vascular endothelial growth factor (VEGF) agents are the first-line treatment for diabetic macular edema (DME), retinal vein occlusion (RVO), myopic choroidal neovascularization (mCNV) and neovascular AMD (nAMD) [12]. The retina is a light-sensitive tissue composed of the RPE and neuroretinal components including photoreceptor cells [13]. The RPE is a distinctive tissue exposed to high oxidative stress caused by irradiation of ambient light in modern life, which generates reactive oxygen species (ROS) that are associated with the development and progression of AMD [14–16]. In the Beaver Dam Eye Study, sunlight exposure was a candidate factor for the development of age-related maculopathy [17]. Therefore, basic research to examine the pathophysiology of ocular disorder with light exposure, especially blue light irradiation [16], and retinal tissue responses exposed to anti-VEGF agents under light irradiation, is needed.” (lines 67-79, in introduction).

Comment 2. Methods and Research Plan: for the therapeutic agents (aflibercept and used, no mention is made of the vehicle(s) for these reagents. For the experiments, it is critical to use the vehicles as controls. The two agents are also different in mode of action against VEGF, and this has not been discussed in the paper (either Methods or Discussion).

 We thank the reviewer for the comment. We corrected the culture without exposure of anti-VEGF-antibody as relative control not containing reagents, and added the appropriate sentence as follows: (1) 0 μg/mL as relative control not containing reagents (line 149, in methods), (2) none as relative control not containing reagents (line 296, in results), (3) “Regarding anti-VEGF antibodies used in this study, aflibercept is a recombinant fusion protein that mimics VEGF receptor-1, -2 and the Fc region of human immunoglobulin G1, and so binds VEGF-A, -B and placental growth factor with very high affinity [54]. In contrast, ranibizumab is comprised of a humanized monoclonal antibody fragment that binds VEGF-A with high affinity, in other words, stops VEGF-A binding to VEGF receptor-1 and -2 [55]. Therefore, it is noted that direct comparisons between the effects of aflibercept and ranibizumab in our study is inappropriate because the reagents of aflibercept are different from those of ranibizumab.” (lines 469-476, in discussion), (4) “In aflibercept- and ranibizumab exposed experiments, no exposure culture which does not include reagents, was set as relative control.” (lines 498-500, in limitation).

Comment 3. The ARPE19 cell line is frequently used as a model but including adult primary human RPE cultures is important; this is noted as a limitation by the authors.

 We thank the reviewer for the comment. We added the appropriate sentence as follows: (1) “In cell morphology, RPE cells contain melanin pigmentation absorbing excess light, and serve a photoprotective role by absorbing radiation and scavenging free radicals and ROS [56–60]. The present study was designed to investigate cytokine profiles of human RPE cells in the condition, which slight cell damage occurred under photooxidative stress by the light, and ARPE-19 cells was used as representative commercial human RPE cells [27]. The RPE cell line ARPE-19 is established from a single individual by Dunn et al. [27] and is widely provided as a dependable alternative to native RPE [28]. However, ARPE-19 cells incompletely appear the replication of phenotypes in native RPE cells, because the cells lose their specialized phenotypes after multiple passages [28]. Melanin pigmentation is a hallmark of native human RPE cells, and an important indicator of phenotypic maturity of the cells in culture [57,61]. However, ARPE-19 cells do not appear evident melanin pigmentations until 4 months incubation [28]. Therefore, the cytokine profiles in light irradiated culture were limitedly obtained from immature ARPE-19 cells without melanin pigmentations.” (lines 477-490, in discussion), (2) “The ARPE19 cell line is used as a human RPE cell model but includes adult primary human RPE cells. Furthermore, the culture conditions adopted in this study was short-term culture, and the cells used were immature ARPE-19 cells without melanin pigmentations.” (lines 502-505, in limitation).

Comment 4. Complete details on how many repeats for each experiment are also required, and the details are not always easy to follow in places.

 We thank the reviewer for the comment. We added the appropriate sentence as follows: (1) “The cells were trypsinized, and 1.5 x 104 cells per well were subcultured in 4-well plates (Thermo Scientific, Nunc, Rochester, NY), in which the area is 1.9 cm2. The cells grown to confluence for 7 days.” (lines 114-116, in methods), (2) “Some fluctuations of obtained cytokine values occurs, even if the experimental conditions are strictly implemented. In this study, preliminary experiments were conducted twice, and the third experimental results were adopted. The results of the preliminary experiments were almost the same with those of the third experiment.” (lines 162-165, in methods)

Comment 5. The initial analysis of the cytokine levels using a principal component approach is confusing and why the detection rates of 80% (of what?), and levels of more than 10pg/ml are used is not well explained. Am also unclear if these criteria were applied for the aflibercept and ranibizumab experiments.

 We thank the reviewer for the comment. We added the appropriate sentence as follows: (1) “To evaluate the comprehensive effects of inflammatory cytokines by hierarchical cluster analysis and PCA, the cytokines used as explanatory variables should be reliably detectable from samples at reasonably high levels. Furthermore, the number of samples must be more than that of one added to number of explanatory variables in these multivariate analyses. On the other hand, IL-6, IL-8, IL-12, IL-17A, bFGF, IP-10, MCP-1 and VEGF had detection rates above 80% and levels higher than 10 pg/mL (Table 1). Therefore, we selected these cytokines as explanatory variables in subsequent analyses.” (lines 207-214, in results), (2) “The levels of IL-6, IL-8, IL-12, L-17A, bFGF, MCP-1 and VEGF were higher than 10 pg/mL in at least one of the 3 cultures, and were selected as cytokines for subsequent analyses.” (lines 296-298, in results).

Comment 6. The correlations between the selected cytokines and growth factors at best, shows association.

 We thank the reviewer for the comment. In this study, the number of experimental cultures exposed to anti-VEGF antibody is four, and too few to analyze the correlations between the selected cytokines and growth factors. Therefore, we could not perform the correlation analysis between the experimental cultures.

Comment 7. The Discussion is very long and not always easy to follow.

 We thank the reviewer for the comment. We deleted the redundant sentence in discussion.

Comment 8. The secreted cytokine and growth factor profile of ARPE19 cells in this study (IL6, -8, -12, -17A, bFGF, IP-10, MCP! and VEGF) is reported, with the conclusion that the primary effect of these is angiogenesis - there are no experiments conducted to support this (although much in the literature).

 We thank the reviewer for the comment. As you pointed out, ARPE-19 cells secret so many cytokines with angiogenesis or anti-angiogenesis, simultaneously. So, the comprehensive effects of cytokine secreted is unclear. In this study, we try to reveal the comprehensive effects using multivariate analyses, including principal component analysis and hierarchical cluster analysis.

Comment 9. The combination of light and the anti-VEGF drugs requires further justification.

 We thank the reviewer for the comment. We added the appropriate sentence as follows: “At present, anti-vascular endothelial growth factor (VEGF) agents are the first-line treatment for diabetic macular edema (DME), retinal vein occlusion (RVO), myopic choroidal neovascularization (mCNV) and neovascular AMD (nAMD) [12]. The retina is a light-sensitive tissue composed of the RPE and neuroretinal components including photoreceptor cells [13]. The RPE is a distinctive tissue exposed to high oxidative stress caused by irradiation of ambient light in modern life, which generates reactive oxygen species (ROS) that are associated with the development and progression of AMD [14–16]. In the Beaver Dam Eye Study, sunlight exposure was a candidate factor for the development of age-related maculopathy [17]. Therefore, basic research to examine the pathophysiology of ocular disorder with light exposure, especially blue light irradiation [16], and retinal tissue responses exposed to anti-VEGF agents under light irradiation, is needed.” (lines 67-79, in introduction).

Comment 10. The authors also discuss melanin in RPE cells (p.16) however ARPE19 cells have very low levels of dark melanin, and the relevance is uncertain.

 We thank the reviewer for the comment. We added the appropriate sentence as follows: (1) “In cell morphology, RPE cells contain melanin pigmentation absorbing excess light, and serve a photoprotective role by absorbing radiation and scavenging free radicals and ROS [56–60]. The present study was designed to investigate cytokine profiles of human RPE cells in the condition, which slight cell damage occurred under photooxidative stress by the light, and ARPE-19 cells was used as representative commercial human RPE cells [27]. The RPE cell line ARPE-19 is established from a single individual by Dunn et al. [27] and is widely provided as a dependable alternative to native RPE [28]. However, ARPE-19 cells incompletely appear the replication of phenotypes in native RPE cells, because the cells lose their specialized phenotypes after multiple passages [28]. Melanin pigmentation is a hallmark of native human RPE cells, and an important indicator of phenotypic maturity of the cells in culture [57,61]. However, ARPE-19 cells do not appear evident melanin pigmentations until 4 months incubation [28]. Therefore, the cytokine profiles in light irradiated culture were limitedly obtained from immature ARPE-19 cells without melanin pigmentations.” (lines 477-490, in discussion), (2) “The ARPE19 cell line is used as a human RPE cell model but includes adult primary human RPE cells. Furthermore, the culture conditions adopted in this study was short-term culture, and the cells used were immature ARPE-19 cells without melanin pigmentations.” (lines 502-505, in limitation).

Comment 11. The complex interplay of cytokines and bFGF are interesting but further investigation is required to identify the modes of action, including for example, expression patterns of receptors for these factors in th retina, RPE and choroid.

 We thank the reviewer for the comment. We added the appropriate sentence as follows: “In the future, further investigations are required to identify the comprehensive modes of cellular responses, including expression patterns of receptors for secreted cytokines by mass cytometry [62] in the retina, RPE and choroid.” (lines 491-493, in discussion).

Comment 12. The Figures showing effects of light exposure and then dark should be combined in the same figure (same graphs, filled and empty bars) to allow easier comparison between conditions. This also applies to the graphs showing concentration effects for the two drugs used under light and dark conditions.

 We thank the reviewer for the comment. We combined Figure 3 and Figure 4 to make Figure 3, Figure 7 and Figure 8 to Figure 6, Figure 9 and Figure10 to Figure 7, respectively.

Reviewer 2 Report

The purpose of the study by Sato and co-authors was to examine 27 inflammatory cytokines secreted by ARPE-19 cells exposed to light irradiation and anti-vascular endothelial growth factor (VEGF) antibody, and evaluated the comprehensive effects of those cytokines using multivariate analyses, aiming to better understand the pathophysiology of ocular diseases related to light irradiation. The results of their study indicate that light irradiation is an angiogenesis-stimulating factor, and inflammatory cytokines work complementarily as angiogenic factors under VEGF suppression.

The study is generally interesting, clearly presented and its topic is in agreement with the purpose of the Journal. I think that the manuscript is well written, referenced, the interpretation of the data supports their conclusions, and deserve consideration for publication.

Minor comments to the Authors:

Abstract section: please, could you rephrase the sentence “In dark cultures, IL-6, IL-8, IL-12, IL-17A, bFGF, IP-10, MCP-1 and VEGF were significant cytokines with levels over 10 pg/mL.”?

Fifth section, page 16: please, change the word “Discussion” with the word “Conclusions”.

Author Response

Responses to the Comments of Reviewers

 We thank the editor and referees for taking their time to review our manuscript. Following Reviewer #2’s suggestions, we have made some corrections and clarifications in the manuscript. All the coauthors have read and agreed with the changes made in the revised manuscript.

 We hope the corrections and revisions are satisfactory, and the revised version will be acceptable for publication. We thank once again Reviewer #2 for the constructive comments. Our responses to the comments and the changes are summarized below.

Responses to Reviewer #2

Suggestion 1: Abstract section: please, could you rephrase the sentence “In dark cultures, IL-6, IL-8, IL-12, IL-17A, bFGF, IP-10, MCP-1 and VEGF were significant cytokines with levels over 10 pg/mL.

 We thank the reviewer for the comment. We corrected the indicated sentence as follows: “In the cells cultured in the dark or under 2000-lux visible-light irradiation for 24 hours, levels of IL-9, IL-17A and bFGF were higher, and levels of IL-6, IL-7, IL-8 and MCP-1 were lower in light irradiated culture than those in dark culture, while there was no significant difference with VEGF level” (lines 39-43, in abstract).

Suggestion 2: Fifth section, page 16: please, change the word “Discussion” with the word “Conclusions”.

 We thank the reviewer for the comment. We corrected the word “Discussion” with the word “Conclusions” (line 507).

Round 2

Reviewer 1 Report

Thank you for the revised manuscript and responses.

Introdcution

"RPE cells regulate the development of immunological and inflammatory responses, and contribute to maintain an immune privileged status by secreting immunosuppressive factors in the posterior segment of the eye."

Surely noting the outer BRB is crtical and must be noted.

DiscussionLine The commentary noted previously on the ant-VEGF agents has now been included in the Discussion (should in Methods as well). Thank you.

Line 475. The issue of negative controls is still not addressed appropriately. For the two VEGF inhibiting therapeutics, the negative control is the vehicle for the drugs (noted before). The vehicle contains components that may affect cell biology. This must be noted as a limitation at the very least. Noting that the controls contained no reagent is not sufficient.

The issue of dark melanin in the ARPE19 cells increasing with time in culture has been published and yes that does eem to be the case. I still do not understand what is the direct relevance to the current study. While light absorption will vary with levels of dark melanin, did the authors check for exmaple on pheomelanin, red-yellow melanin and considered pro-oxidative stress (compared to dark melanin).  

Line 479. 'The ARPE19 cell line is used as a human RPE cell model but includes adult primary human RPE cells.'

A celi line does not include primary cultures of human RPE cells - what does this mean?